

# CurriMAE: curriculum learning based masked autoencoders for multi-labeled pediatric thoracic disease classification

Taeyoung Yoon and Daesung Kang

School of Bio-Health Convergence, College of Natural Sciences, Sungshin Women's University, Seoul, Republic of Korea

## ABSTRACT

Masked autoencoders (MAE) have emerged as a powerful framework for self-supervised learning by reconstructing masked input data. However, determining the optimal masking ratio requires extensive experimentation, resulting in significant computational overhead. To address this challenge, we propose CurriMAE, a curriculum-based training approach that progressively increases the masking ratio during pretraining to balance task complexity and computational efficiency. In CurriMAE, the training process spans 800 epochs, with the masking ratio gradually increasing in four stages: 60% for the first 200 epochs, followed by 70%, 80%, and finally 90% in the last 200 epochs. This progressive masking approach, inspired by curriculum learning, allows the model to learn from simpler tasks before tackling more challenging ones. To ensure stable convergence, a cyclic cosine learning rate scheduler is employed, resetting every 200 epochs, effectively dividing the training process into four distinct stages. At the end of each stage, corresponding to one complete cycle of the learning rate schedule, a snapshot model is saved, resulting in four pretrained models. These snapshots are then fine-tuned to obtain the final classification results. We evaluate CurriMAE on multi-labeled pediatric thoracic disease classification, pretraining the model on CheXpert and ChestX-ray14 datasets, and fine-tuning it on PediCXR. Experimental results show that CurriMAE outperforms ResNet, ViT-S, and standard MAE, achieving superior performance while reducing computational cost. These findings establish CurriMAE as an effective and scalable self-supervised learning framework for medical imaging applications.

## INTRODUCTION

The automated detection of various chest-related disorders in children's X-ray images is a critical area of medical research. Respiratory illnesses such as pneumonia, bronchitis, and bronchiolitis rank among the primary reasons for illness and death in young patients, especially in areas with limited healthcare resources (*Unicef and World Health Organization, 2006*). Timely and precise diagnosis plays a vital role in starting effective treatments and preventing serious health issues.

Corresponding author
Daesung Kang,
danniskang@gmail.com

In recent times, deep learning techniques, particularly convolutional neural networks (CNNs), have shown great potential in this field by delivering impressive results in analyzing chest X-ray images (*Irvin et al., 2019*; *Wang et al., 2017*). Research involving specific pediatric datasets has confirmed the possibility of identifying multiple chest conditions simultaneously through supervised learning methods (*Pham et al., 2023*). However, these approaches often demand extensive labeled datasets, which are hard to obtain for pediatric cases due to ethical issues, high costs of data labeling, and the relative rarity of certain diseases. Moreover, many existing models are pretrained on ImageNet, which do not adequately reflect the unique anatomical and pathological characteristics present in pediatric chest X-rays. This discrepancy can result in reduced accuracy and challenges in applying these models across diverse medical environments. To tackle such challenges, recent studies have explored self-supervised learning (SSL) approaches, which mitigate the need for labeled data by extracting patterns directly from the data itself (*Chen et al., 2020*; *He et al., 2022*, *2020*).

SSL has gained significant attention in recent years, particularly for its ability to learn useful representations from unlabeled data. For example, SSL methods have been successfully applied in natural language processing to pretrain language models, such as bidirectional encoder representations from transformers (BERT), by predicting masked words in a sentence, enabling them to capture contextual relationships (*Devlin et al., 2019*; *Huang et al., 2017*). In computer vision, SSL has been widely utilized to learn feature representations without requiring extensive labeled datasets. Methods such as contrastive learning (*e.g.*, SimCLR, MoCo) leverage instance discrimination to distinguish positive and negative pairs, while predictive modeling approaches, including masked image modeling (MIM), reconstruct missing parts of an image to enhance feature learning (*Bao et al., 2022*; *Chen et al., 2020*; *He et al., 2020*). These techniques have demonstrated strong performance in various tasks by enabling models to capture hierarchical and semantic structures from image patches. These methods significantly reduce the dependency on large labeled datasets, making them especially valuable in fields like medical imaging, where annotating data is often time-consuming and requires domain expertise (*Bozorgtabar, Mahapatra & Thiran, 2023*; *Cai et al., 2022*; *Wang et al., 2024*).

Among the various SSL techniques, masked autoencoders (MAE) have shown remarkable performance in multiple domains, including medical imaging and computer vision. By masking portions of the input data and reconstructing the missing parts, MAE effectively learns meaningful features in an unsupervised manner. For instance, in medical imaging, MAE can be applied to reconstruct missing regions of X-ray, helping the model focus on critical features such as anatomical structures or pathological markers (*Saeed Shurrab, 2022*; *Xiao et al., 2023*; *Yoon & Kang, 2024b*). In computer vision, MAE has been used to pretrain vision transformers by reconstructing pixel-level details of masked image patches, enabling robust feature extraction for downstream tasks like image classification, object detection, and segmentation (*Dai et al., 2023*; *He et al., 2022*; *Xie et al., 2023*). These capabilities allow MAE to generalize across diverse datasets, making it a powerful tool for high-dimensional and complex data scenarios.

Despite their promise, MAE models face a critical challenge: the computational burden associated with identifying an optimal masking ratio. Determining the best masking ratio typically requires extensive experimentation, which involves training multiple models with different masking ratios. This process is not only resource-intensive but also time-consuming for large-scale datasets and high-dimensional tasks. To address this limitation, we propose CurriMAE, a novel curriculum-based training strategy for MAE. Inspired by the principles of curriculum learning (*Bengio et al., 2009*), where tasks are structured in increasing order of difficulty, CurriMAE introduces a progressive masking ratio schedule during pretraining. Instead of fixing a masking ratio or exhaustively searching for an optimal masking ratio, CurriMAE begins with simpler tasks (lower masking ratios) and gradually transitions to more challenging ones (higher masking ratios). That is, CurriMAE pushes the model to handle highly challenging tasks with minimal input visibility. This approach enables efficient learning while alleviating the computational complexity associated with standard MAE. To further optimize the learning process, we employ a cyclic cosine learning rate scheduler (*Huang et al., 2017*), which helps the model converge efficiently at each phase.

The contributions of this research are summarized as follows:

- We address the computational inefficiencies of MAE by eliminating the need for exhaustive masking ratio tuning through a progressive masking strategy.
- We propose CurriMAE, a curriculum-based training framework that gradually increases the masking ratio during pretraining, effectively balancing task difficulty and computational efficiency.
- We evaluate CurriMAE on multi-labeled pediatric thoracic disease classification, demonstrating superior performance over ResNet, ViT-S, and standard MAE, while maintaining computational efficiency.
- We compare fixed-epoch and adaptive-epoch training strategies, showing that progressively allocating training epochs based on task complexity can further enhance model performance.

## RELATED WORKS

Pediatric thoracic disease classification has recently gained attention due to the clinical importance of early detection of respiratory illnesses in children. Several studies have employed deep learning approaches on pediatric chest X-rays such as PediCXR (*Pham et al., 2023*) and shown significant advancements through the integration of large-scale datasets, deep learning, and refined clinical guidelines. *Chen et al. (2021)* developed a deep learning model for classifying pediatric chest X-rays based on World Health Organization's standardized methodology. The model utilized transfer learning from a large adult CXR dataset and was fine-tuned on a pediatric dataset. The model achieved performance comparable to or exceeding that of radiologists and pediatricians, particularly excelling in cases with high inter-observer agreement, and demonstrated the potential for automated, standardized interpretation of pediatric CXRs in clinical and epidemiological

studies. *Tran et al. (2021)* developed a CNN framework for multi-label classification of pediatric chest radiographs, addressing challenges such as limited annotated datasets and severe class imbalance by introducing a modified distribution-balanced loss function. Their approach, validated on a large, expert-annotated pediatric CXR dataset, outperformed previous state-of-the-art methods for detecting ten common thoracic diseases and demonstrated the effectiveness of tailored loss functions in improving diagnostic performance on imbalanced pediatric data. *Yoon & Kang (2024a)* proposed a dual-masked autoencoder (dual-MAE) framework for multi-label classification of pediatric thoracic diseases, which leverages pretraining on large adult chest X-ray datasets followed by fine-tuning on pediatric data to overcome the challenge of limited pediatric samples. Their dual-MAE model outperformed conventional architectures and other pretraining strategies, achieving the highest mean AUC in multi-label pediatric thoracic disease classification, and demonstrated robust performance even with only half of the labeled pediatric data available.

Curriculum learning has long been a foundational concept in machine learning, emphasizing the benefits of structured learning progression from simpler to more complex tasks. The curriculum-based methodology aligns closely with human cognitive learning processes, providing a foundation for more bio-inspired machine learning paradigms (*Azimi et al., 2021*). *Bengio et al. (2009)* introduced curriculum learning by formalizing the idea that training machine learning models with examples organized in increasing order of difficulty can improve both generalization and convergence speed. They hypothesized that curriculum learning operates as a continuation method, where simpler tasks help guide optimization toward better local minima of non-convex training objectives. Through experiments in tasks like shape recognition and language modeling, the authors demonstrated that curriculum learning not only accelerates convergence but also improves generalization by reducing the learner's initial exposure to noisy or overly complex examples. These findings present the adaptive advantage of gradually increasing task complexity during training. *Tan & Le (2021)* introduced EfficientNetV2, which incorporated various techniques to enhance training efficiency and accuracy. Among them, a curriculum learning-inspired progressive learning approach was applied to optimize the training process. The training began with smaller image sizes and weaker regularization, allowing the model to quickly learn simple representations. As training progressed, image sizes were gradually increased, and stronger regularization techniques, such as dropout, data augmentation, and mixup, were introduced to increase complexity. This gradual adjustment aligns with the principles of curriculum learning, where models first learn easier tasks before progressing to more difficult ones, facilitating better optimization and improved generalization. Experiments on ImageNet demonstrated that this progressive learning strategy not only accelerated training but also maintained or improved accuracy compared to previous methods.

Masked image modeling (MIM) has emerged as a powerful SSL paradigm, particularly in vision tasks. Vision Transformers (ViT) paved the way for MIM approaches by demonstrating the effectiveness of patch-based input representations (*Dosovitskiy et al., 2021*). *Bao et al. (2022)* introduced Bidirectional Encoder representation from Image

Transformers (BEiT), which established MIM as a self-supervised pretraining task for ViT, inspired by BERT's masked language modeling in NLP. In BEiT, images were represented as discrete visual tokens generated by a tokenizer based on a discrete variational autoencoder (dVAE). During pretraining, BEiT randomly masked a portion of image patches and learned to predict the corresponding visual tokens instead of reconstructing raw pixels. This approach addressed the limitations of pixel-level autoencoding by focusing on high-level abstractions rather than short-range dependencies. BEiT demonstrated strong performance on downstream tasks such as image classification and semantic segmentation, emphasizing the effectiveness of MIM in learning robust visual representations. *Xie et al. (2022)* proposed SimMIM, a simple yet effective framework for MIM that learns visual representations by reconstructing raw pixel values in masked image patches. Unlike previous methods that relied on complex mechanisms such as tokenization *via* dVAEs (*Bao et al., 2022*) or clustering (*Caron et al., 2018*), SimMIM employed a straightforward approach: random patch masking, a lightweight linear prediction head, and pixel regression with an $l_1$ loss. This method demonstrated that raw pixel regression aligned effectively with the continuous nature of visual signals, enabling the model to achieve competitive results without requiring intricate pretext tasks. Despite its simplicity, SimMIM outperformed more complex methods on benchmarks such as ImageNet-1K and scaled effectively to larger models like SwinV2-H, highlighting the potential of MIM as an efficient and robust self-supervised learning strategy.

Recent research has explored the integration of curriculum learning with MIM to enhance SSL frameworks. *Madan et al. (2024)* introduced curriculum-learned masked autoencoders (CL-MAE), CL-MAE integrated curriculum learning with MIM to enhance self-supervised representation learning. The approach employed a novel learnable masking module that generated masks of increasing complexity throughout training. Initially, the module produced easy masks to simplify the reconstruction task, gradually transitioning to harder masks by dynamically adjusting a curriculum loss factor. This easy-to-hard progression allowed the MAE to progressively adapt to more challenging reconstruction tasks, leading to better generalization and robust feature learning. Experimental results demonstrated that this curriculum-based masking strategy significantly improved representation quality compared to standard MIM techniques, validating the synergy between curriculum learning and MIM. *Lin et al. (2024)* proposed a prototype-driven curriculum learning framework for MIM to address optimization challenges during early training stages. Instead of exposing the model to the full complexity of natural image distributions from the start, the approach began with prototypical examples representing essential visual patterns for each semantic category. As training progresses, the curriculum gradually incorporated more complex variations, enabling stable and efficient learning trajectories. The proposed framework improved training efficiency and representation quality, achieving superior performance on ImageNet-1K compared to standard MAE training strategies. This demonstrated that curriculum learning principles can effectively enhance MIM by structuring the learning process from simple to complex examples.

Table 1 summarizes the differences between CurriMAE and recent curriculum-based MIM approaches.

**Table 1  Comparison of CurriMAE with existing curriculum-based MIM methods.**

| Method | CL-MAE (*Madan et al., 2024*) | Prototype-CL (*Lin et al., 2024*) | CurriMAE (ours) |
|---|---|---|---|
| Curriculum strategy | Dynamic/self-paced | Fixed, data-driven | Fixed-stage |
| Curriculum target | Mask complexity | Data semantic complexity | Masking ratio |
| Curriculum progression | Learnable mask complexity | From prototypes to general samples | Predefined (60% to 90%) |
| Masking strategy | Learnable masking module | Sampling based on clustering + annealing | Random masking with progressive difficulty |
| Pretraining snapshots | No | No | Yes (4 snapshots) |
| Fine-tuning required | No | No | Yes (per snapshot) |
| Application domain | Natural image benchmarks | ImageNet-1K | Pediatric thoracic disease |

# MATERIALS AND METHODS

## Datasets

In this study, we evaluated CurriMAE on a multi-labeled pediatric chest X-ray dataset. Due to the limited size of the PediCXR dataset (*Pham et al., 2023*), CurriMAE was not pretrained directly on PediCXR. Instead, pretraining was conducted on the CheXpert dataset (*Irvin et al., 2019*) and ChestX-ray14 dataset (*Wang et al., 2017*), which contain adult chest X-rays that share structural, morphological, and textural similarities with pediatric chest X-rays (*Yoon & Kang, 2024b*). The overall experimental pipeline involved pretraining CurriMAE on the CheXpert and ChestX-ray14 datasets, followed by fine-tuning on the PediCXR dataset, which comprises multi-labeled pediatric thoracic disease cases.

The CheXpert dataset consists of 224,316 chest radiographs, including 191,229 frontal view images and 33,087 lateral view images. These radiographs are categorized into fourteen classes, covering twelve pathologies, the presence of medical support devices, and normal cases. In this study, only the 191,229 frontal view images were utilized, while lateral view images were excluded. Similarly, the ChestX-ray14 dataset comprises 112,120 frontal chest X-ray images, among which 51,708 are annotated with at least one pathology spanning fourteen categories, while 60,412 represent normal cases. In total, 303,349 images from the CheXpert and ChestX-ray14 datasets were used for pretraining, as summarized in Table 2.

After pretraining, the PediCXR dataset was used to fine-tune the models for pediatric thoracic disease classification. This dataset contains 9,125 pediatric chest X-ray images, officially divided into 7,728 training images and 1,398 test images. The training set originally included fifteen disease classes, but four diseases—congenital emphysema, diaphragmatic hernia, lung tumor, and pleuro-pneumonia—were absent in the test set. To maintain consistency, these diseases were merged into the "other diseases" class. Additionally, five disease classes—CPAM, hyaline membrane disease, mediastinal tumor, situs inversus, and tuberculosis—had fewer than twenty training samples each. These were also incorporated into the "other diseases" class. Consequently, the PediCXR dataset was restructured into six final classes: no finding, bronchitis, broncho-pneumonia, bronchiolitis, pneumonia, and other diseases, as detailed in Table 2.

**Table 2 Overview of datasets used for pretraining and fine-tuning in multi-labeled pediatric thoracic disease classification, including the number of samples for each class.**

| Data | Dataset | Class | Pretraining samples | Test samples |
|---|---|---|---|---|
| Pretraining data | CheXpert | – | 191,229 | – |
| | ChestX-ray14 | – | 112,120 | – |
| Fine-tuning data | PediCXR | No finding | 5,143 | 907 |
| | | Bronchitis | 842 | 174 |
| | | Broncho-pneumonia | 545 | 84 |
| | | Bronchiolitis | 497 | 90 |
| | | Pneumonia | 392 | 89 |
| | | Other diseases | 485 | 85 |

For model fine-tuning and training from scratch with randomly initialized weights, 80% of the training data was used for training, while the remaining 20% was allocated for validation. Given that chest X-rays often exhibit multiple co-occurring medical conditions, this classification task was treated as a multi-labeled problem, allowing each image to be associated with multiple disease labels. The samples used for pretraining on adult chest X-rays (first row) and fine-tuning on pediatric chest X-rays (second row) are illustrated in Fig. 1.

The PediCXR dataset includes chest radiographs from patients younger than ten years. However, detailed metadata such ICU admission status, and disease severity levels are not provided in the public release of the dataset.

Ethical review and approval were not required, as the study exclusively analyzed anonymized clinical open data.

## Pretraining CurriMAE model

The CurriMAE framework is designed to mitigate the computational inefficiencies associated with standard MAE training by incorporating a progressive masking ratio schedule alongside a cyclic cosine learning rate scheduler. The core methodology consists of pretraining the MAE model on unlabeled data, followed by fine-tuning the pretrained encoder, specifically the ViT-S model, using labeled data as depicted in Fig. 2. This approach was applied to the multi-labeled pediatric thoracic disease task.

As illustrated in Fig. 2A, CurriMAE began by dividing input images into non-overlapping $16 \times 16$ patches. Each patch was flattened, projected into a low-dimensional token *via* linear projection, and added with positional embeddings. A subset of these tokens was randomly selected based on predefined masking ratios, and the selected tokens were masked.

To enhance representation learning, CurriMAE employs a progressive masking strategy over 800 epochs. Training started with a 60% masking ratio for the first 200 epochs, allowing the model to focus on relatively simpler reconstruction tasks. At the end of this phase, the model was saved as a snapshot. The masking ratio was then gradually increased to 70%, 80% and 90% over subsequent 200-epoch intervals, progressively increasing the task complexity. At the end of each phase, corresponding to every 200 epochs, model

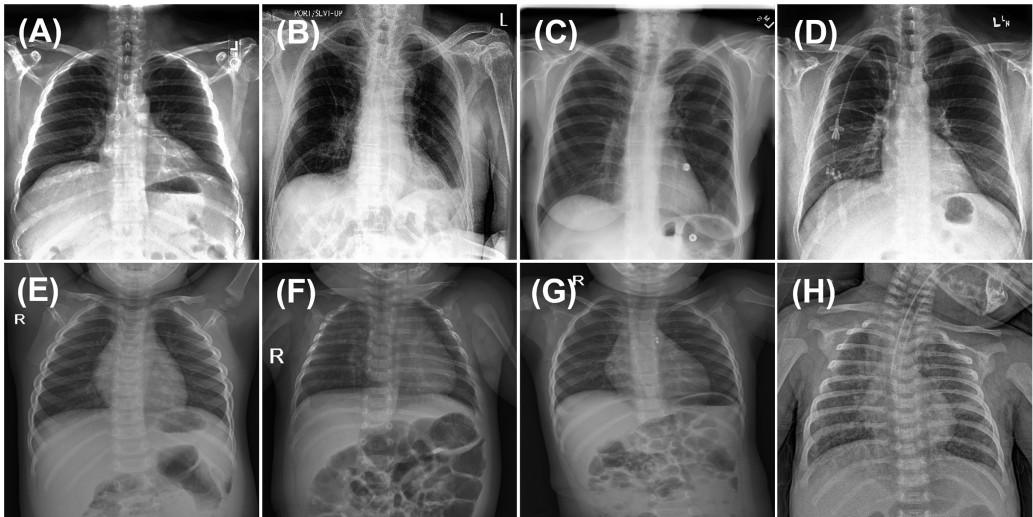

**Figure 1 Sample images from CheXpert and ChestX-ray14 datasets (first row) and PediCXR dataset (second row).** Each image may present multiple conditions: (A) No finding. (B) Atelectasis & Pleural Effusion. (C) Effusion & Infiltration. (D) Lung Opacity & Atelectasis. (E) No finding. (F) Bronchiolitis & Pneumonia. (G) Other disease and (H) Pneumonia & Other disease.

snapshots were saved and later used for fine-tuning. This curriculum learning approach gradually exposes the model to increasingly challenging tasks, enhancing representation learning and improving generalization capabilities.

To complement the progressive masking approach, CurriMAE employed a cyclic cosine learning rate scheduler instead of conventional monotonic schedulers. The cyclic cosine scheduler dynamically oscillated the learning rate between a minimum and maximum value following a cosine function, enabling more effective exploration of the parameter space (*Huang et al., 2017*). The learning rate schedule is defined as in Eq. (1):

$$\alpha(t) = \frac{\alpha_0}{2}\left(cos\left(\frac{\pi \mathrm{mod}(t-1, \lceil T/M \rceil)}{\lceil T/M \rceil}\right) + 1\right), \tag{1}$$

where $\alpha_0$ represents the initial learning rate, $t$ is the iteration number, $T$ denotes the total number of training iterations, and $M$ is the number of cycles. This study employed a training regimen of 800 epochs with an initial learning rate ($\alpha_0$) of 1.5e−4, number of cycles ($M$) of 4. This approach periodically increases the learning rate, allowing the model to escape shallow local minima and explore alternative optimization paths, resulting in improved convergence speed and stability.

For each masking ratio, the unmasked tokens were processed by a Transformer encoder, which comprised multi-head self-attention layers, feed-forward neural networks, layer normalization, and residual connections, ensuring stable training. The encoder extracts global representations from partially visible tokens. These encoded tokens were then passed to the decoder, where they were combined with learnable masked tokens, applied positional embeddings, and used to reconstruct the missing patches. The decoder then
**(A) Pretraining**

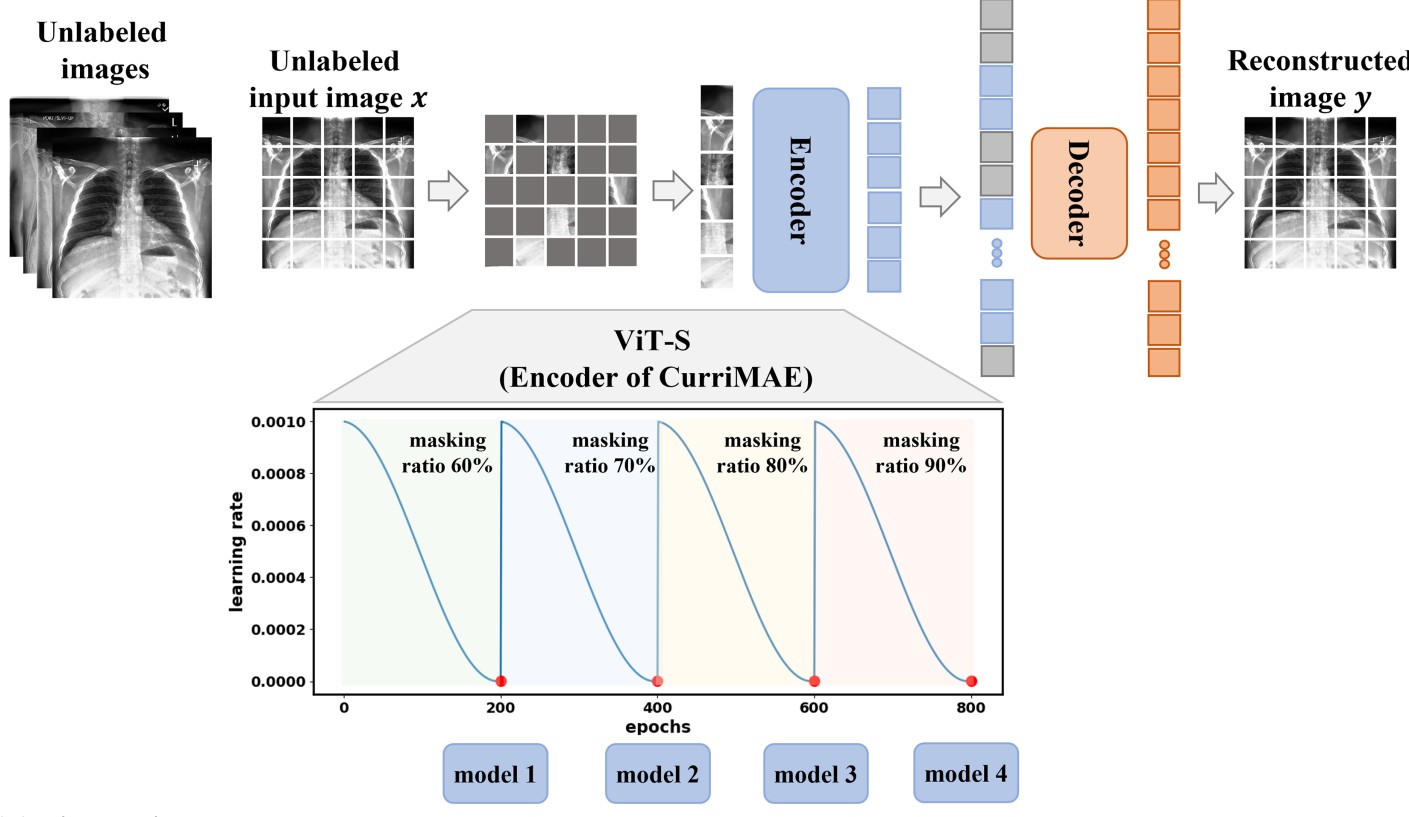

**(B) Fine-tuning**

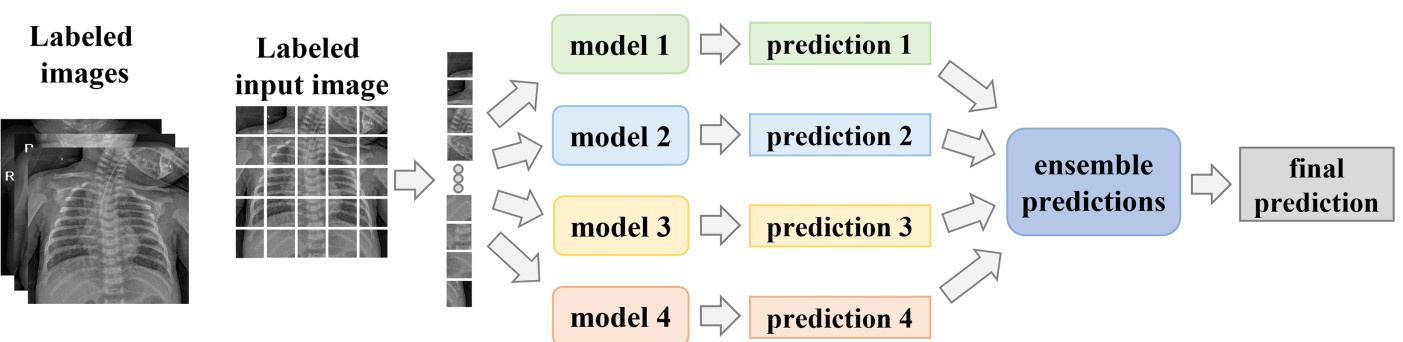

**Figure 2 CurriMAE framework.** (A) Pretraining phase using CheXpert and ChestX-ray14 datasets. (B) Fine-tuning phase using the PediCXR dataset.

reshaped the output into a fully reconstructed image, leveraging both visible and masked tokens to restore the missing patches.

CurriMAE was trained to minimize the mean squared error (MSE) loss between the original unlabeled images ($X$) and their reconstructions ($Y$), focusing exclusively on the

masked patches. Given an input image divided into non-overlapping patches ($X_i$), with masked blocks ($B$), the MSE loss was formulated as in Eq. (2):

$$MSE\ loss = \frac{1}{|B|} \sum_{i \in B} ||X_i - Y_i||_2^2. \tag{2}$$

The AdamW optimizer was used to minimize the MSE loss, with parameters $\beta_1 = 0.9$, $\beta_2 = 0.99$, and a weight decay of 0.05. During pretraining, weak data augmentations such as resizing, cropping, and horizontal flipping were applied to prevent bias in clinically significant lesions or anatomical structures. The images were first resized to $256 \times 256$ pixels, then cropped to $224 \times 224$ pixels, and standardized using ImageNet dataset statistics (mean and standard deviation). The Transformer blocks in CurriMAE were initialized using Xavier uniform initialization and the batch size was 256. Pretraining was conducted over 800 epochs using a progressive masking ratio schedule in conjunction with a cyclic cosine learning rate scheduler. This framework effectively integrates curriculum learning principles into MAE, allowing CurriMAE to learn robust representations while maintaining computational efficiency. The pretrained models can then be fine-tuned using labeled data, enabling adaptation to various downstream tasks and demonstrating versatility across diverse domains.

### Fine-tuning ViT-S network

In this study, the pretrained encoder of the CurriMAE model is based on the ViT-S network, which is fine-tuned for downstream tasks using labeled data in an end-to-end manner, as illustrated in Fig. 2B. After the pretraining phase, models saved at every 200 epochs were utilized for fine-tuning. Similar to the pretraining stage, input images were divided into $16 \times 16$ non-overlapping patches. However, during fine-tuning, all patches were retained without masking. Each patch was flattened, projected into an embedding space through linear projection, and combined with positional embeddings to maintain spatial information. These embeddings were then processed by the pretrained Transformer encoder. Following the methodology in *Dosovitskiy et al. (2021)*, a linear classifier was attached to the class token output of the ViT-S network to obtain predictions.

For the pediatric thoracic disease classification task, a multi-labeled classification approach was employed, allowing each input image to be assigned multiple disease labels, as illustrated in Fig. 1. To accommodate this, the final fully connected layer of the ViT-S network was adjusted to generate a six-dimensional output with an elementwise sigmoid activation, representing the probability of each disease. The model was trained using the mean of binary cross-entropy (BCE) losses to effectively handle the multi-labeled classification setting (*Chen, Bai & Zhang, 2024*; *Zhang & Zhou, 2013*). Predicted probabilities were obtained through sigmoid functions. For robust and stable performance, an ensemble method was employed, where predictions from all snapshot models were averaged, as expressed in Eq. (3):

$$h_{ensemble} = \frac{1}{m} \sum_{j=1}^{M} h_j(x), \tag{3}$$

where $x$ represents a test sample, and $h_j(x)$ denotes the sigmoid score from the $j^{th}$ snapshot model. In multi-labeled classification, a threshold was applied to determine whether a class was present. This ensemble model, $h_{ensemble}$, leverages the strengths of multiple models, leading to improved overall performance (*Benzorgat, Xia & Benzorgat, 2024*).

For pediatric thoracic disease classification, the ViT-S network was trained using the AdamW optimizer with parameters $\beta_1 = 0.9$, $\beta_2 = 0.99$, and a weight decay of 0.05. The initial learning rate was set to 2.5e−3 and adjusted using a cosine scheduler, with a batch size of 128. Fine-tuning was conducted over 75 epochs, including a 5-epoch warm-up period. To enhance training stability and performance, a layer-wise learning rate decay of 0.55, a RandAug magnitude of 6, and a DropPath rate of 0.2 were utilized, following recommendations from *Xiao et al. (2023)*. To ensure robustness, each experiment was repeated three times using different random seeds for weight initialization. The pretraining and fine-tuning hyperparameters for CurriMAE are summarized in Table 3. The code for CurriMAE is available at: https://github.com/xodud5654/CurriMAE.

## Performance measures

To assess the performance of our diagnostic models, we measured area under the curve (AUC), sensitivity, precision, and F1-score, while accuracy was excluded for specific cases. In the pediatric thoracic disease classification task, the dataset exhibited severe class imbalance, with the "no finding" class accounting for approximately 65% of the samples. Given this imbalance, accuracy was deemed an unsuitable metric and was therefore excluded from the evaluation for this task.

Since pediatric thoracic disease classification is a multiclass problem, AUC was computed separately for each class. To ensure a balanced evaluation across all classes, weighted averaging was applied to AUC, sensitivity, precision, and F1-score. This technique is particularly effective for imbalanced datasets, as it assigns weights to each class based on its relative frequency, preventing dominant classes from disproportionately influencing the overall performance (*Grandini, Bagli & Visani, 2020*). The weighted averaging metric is calculated as following Eq. (4):

$$weighted\ averaging\ metrics = \frac{\sum_{i=1}^{N}(w_i \times m_i)}{\sum_{i=1}^{N} w_i}, \tag{4}$$

where $N$ represents the total number of classes, $w_i$ is the weight assigned to class $i$ (typically the number of samples in that class), and $m_i$ denotes the performance metric for class $i$, such as sensitivity, precision, and F1-score.

## Implementation details and computational analysis

We systematically evaluate the computational requirements and training efficiency of CurriMAE while providing full implementation specifications to ensure reproducibility.

All experiments were conducted on a Linux workstation (Ubuntu 24.04) with an Intel Core Ultra 7 265KF CPU, 128 GB RAM, and an NVIDIA RTX 5090 GPU (32 GB VRAM). The framework was implemented using PyTorch 2.7.0 with CUDA 12.8 acceleration and

**Table 3 Summary of CurriMAE pretraining and fine-tuning hyperparameters.**

| Phase | Parameters | Value |
|---|---|---|
| Pretraining | Epochs | 800 |
| | Masking ratios | 60%, 70%, 80%, 90% (200 epochs each) |
| | Learning rate ($\alpha_0$) | 1.5e−4 |
| | Scheduler | Cyclic cosine learning rate scheduler |
| | Optimizer | AdamW ($\beta_1 = 0.9$, $\beta_2 = 0.99$, weight decay = 0.05) |
| | Batch size | 256 |
| | Augmentation | Resize to 256 × 256, crop to 224 × 224, flip |
| Fine-tuning | Epochs | 75 (5 warm-up) |
| | Learning rate | 2.5e−3 |
| | Optimizer | AdamW ($\beta_1 = 0.9$, $\beta_2 = 0.99$, weight decay = 0.05) |
| | Batch size | 128 |
| | Layer-wise learning rate decay | 0.55 |
| | RandAug | 6 |
| | DropPath | 0.2 |

timm 0.3.2 for model components. Full hardware/software specifications are detailed in Table S1.

To support the claim that CurriMAE reduces computational burden compared to training multiple separate MAE models with different masking ratios, we profiled four architectures under identical conditions: ResNet-34, ViT-S, standard MAE, and CurriMAE. Key profiling metrics include the number of parameters, FLOPs per sample, model size, maximum GPU memory usage, and average training time per epoch. Table 4 summarized these results. As shown in Table 4, CurriMAE maintains the same architectural structure and computational characteristics as standard MAE. However, it eliminates the need to retrain multiple MAE models with different masking ratios, instead generating intermediate snapshot models within a single, continuous pretraining cycle. While fine-tuning is subsequently required for each snapshot model, this step is significantly less expensive compared to pretraining multiple full MAE models, as the fine-tuning process involves a small dataset (7.7K images for 75 epochs) compared to pretraining phase (303K images for 800 epochs per masking ratio). Thus, CurriMAE achieves a favorable balance between performance and training efficiency.

## RESULTS

For the multi-labeled pediatric thoracic disease classification task, a comparative analysis was conducted to evaluate the effectiveness of the proposed CurriMAE framework. The models compared with CurriMAE include MAE, ViT-S, and ResNet-34. Among various model architectures, ViT-S was chosen as a benchmark due to its identical encoder framework to both MAE and CurriMAE, ensuring a fair and relevant comparison. ResNet-34 was included as a representative CNN model, given that its parameter size (approximately 22 million parameters) is comparable to ViT-S (He et al., 2016). The MAE model was also evaluated against CurriMAE to directly assess the impact of progressive

**Table 4 Comparative profiling metrics for different models.**

| Metric | ResNet-34 | ViT-S | MAE | CurriMAE |
|---|---|---|---|---|
| FLOPs per sample (G) | 7.36 | 6.44 | 1.78 | 1.78 |
| Parameters (Millions) | 21.30 | 21.67 | 22.14 | 22.14 |
| Model Size (MB) | 81.37 | 82.73 | 84.89 | 84.89 |
| Max GPU Memory Usage (MB) | 2,512.33 | 1,149.13 | 1,639.89 | 1,639.89 |
| Training time per epoch (min:sec) | 2:29 | 3:55 | 4:36 | 4:30 |

masking and curriculum learning strategies employed during pretraining. Additionally, the final CurriMAE results were obtained by applying a simple averaging ensemble to the four snapshot models.

The terms "random," "IN," and "X-ray" appended to the model names indicate the pretraining methods used. Specifically, "random" refers to models trained from scratch with randomly initialized weights on the CheXpert and ChestX-ray14 datasets, "IN" denotes pretraining on ImageNet, and "X-ray" indicates pretraining performed directly on X-ray datasets. This nomenclature facilitates a clear distinction between different pretraining strategies.

Table 5 presents the performance metrics for the multi-labeled pediatric thoracic disease classification task, including AUC, sensitivity, precision, and F1-score, across various models and pretraining approaches.

Among all models, CurriMAE (X-ray) achieved the highest performance, with an AUC of 0.756, sensitivity of 0.759, precision of 0.556, and F1-score of 0.622, emphasizing its effectiveness in handling multi-labeled classification tasks with imbalanced data. Among the MAE models, the 60% masking ratio delivered strong results, achieving an AUC of 0.746, sensitivity of 0.747, precision of 0.550, and F1-score of 0.609, demonstrating the robustness of progressive masking strategies. ResNet-34 (X-ray) outperformed both ResNet-34 (FS) and ResNet-34 (IN), with an AUC of 0.712, sensitivity of 0.746, precision of 0.520, and F1-score of 0.599. Similarly, ViT-S (X-ray) achieved an AUC of 0.669, outperforming its ViT-S (FS) and ViT-S (IN), yet still trailing behind CurriMAE and MAE models. Models trained from scratch, such as ResNet-34 (FS) and ViT-S (FS), exhibited lower performance, with ResNet-34 (FS) achieving an AUC of 0.675 and ViT-S (FS) scoring 0.648. These findings validate the superiority of CurriMAE over ResNet-34, and ViT-S, showing its effectiveness in multi-labeled pediatric thoracic disease task.

In previous CurriMAE pretraining experiments, the masking ratio was adjusted progressively every 200 epochs (60%, 70%, 80%, 90%), with the model being saved at each stage and the cyclic cosine learning rate being reset. This uniform pretraining schedule, referred to as the fixed-epoch approach, applies the same number of epochs (200) to each masking ratio stage, regardless of task difficulty. However, since reconstructing masked patches at a 60% masking ratio is inherently easier than at 90%, we explored an adaptive epoch strategy that assigns fewer epochs to easier tasks and more epochs to harder tasks.

To test this hypothesis, we designed an incremental pretraining schedule, where the model was trained for 125 epochs at 60% masking, 300 epochs at 70%, 525 epochs at 80%,

**Table 5 Performance comparison of various models for multi-labeled pediatric thoracic disease classification (highest values are bolded; standard deviations are in parentheses).**

| | AUC | Sensitivity | Precision | F1-score |
|---|---|---|---|---|
| ResNet-34 (FS) | 0.675 (0.005) | 0.680 (0.012) | 0.510 (0.002) | 0.560 (0.011) |
| ViT-S (FS) | 0.648 (0.006) | 0.649 (0.009) | 0.504 (0.009) | 0.539 (0.009) |
| ResNet-34 (IN) | 0.698 (0.048) | 0.698 (0.023) | 0.518 (0.013) | 0.591 (0.016) |
| ViT-S (IN) | 0.643 (0.001) | 0.674 (0.022) | 0.489 (0.003) | 0.559 (0.008) |
| ResNet-34 (X-ray) | 0.712 (0.023) | 0.746 (0.008) | 0.520 (0.010) | 0.599 (0.018) |
| ViT-S (X-ray) | 0.669 (0.005) | 0.674 (0.014) | 0.510 (0.010) | 0.557 (0.007) |
| MAE (X-ray) 60% | 0.746 (0.005) | 0.747 (0.009) | 0.550 (0.006) | 0.609 (0.008) |
| MAE (X-ray) 70% | 0.685 (0.053) | 0.696 (0.035) | 0.516 (0.028) | 0.572 (0.027) |
| MAE (X-ray) 80% | 0.737 (0.017) | 0.723 (0.004) | 0.547 (0.011) | 0.593 (0.003) |
| MAE (X-ray) 90% | 0.741 (0.009) | 0.734 (0.006) | 0.550 (0.009) | 0.601 (0.003) |
| CurriMAE (X-ray) | **0.756 (0.002)** | **0.759 (0.005)** | **0.556 (0.004)** | **0.622 (0.006)** |

and 800 epochs at 90%, progressively increasing the number of epochs by 50 at each stage. This incremental strategy was then compared to the fixed-epoch approach (200, 400, 600, 800 epochs per stage) to evaluate its impact on model performance.

As illustrated in Fig. 3, the fixed-epoch approach generally outperformed the adaptive-epoch approach across most masking ratios in terms of AUC, sensitivity, precision, and F1-score. This suggests that allocating an equal number of training epochs per stage allows for more stable learning in earlier phases, where the reconstruction task is relatively straightforward. However, at a 90% masking ratio, the adaptive-epoch approach exhibited higher performance across all metrics, indicating that allocating more training epochs to harder tasks helps the model learn meaningful representations despite having significantly less visible information. This finding suggests that a hybrid approach, optimizing epoch allocation based on task complexity, could further enhance CurriMAE's performance.

To further assess the effectiveness of these approaches, we compared their ensemble results by aggregating model snapshots from all stages. The ensemble from the fixed-epoch approach (200, 400, 600, and 800 epochs) achieved an AUC of 0.756, sensitivity of 0.759, precision of 0.556, and F1-score of 0.622. In contrast, the adaptive-epoch ensemble (125, 300, 525, and 800 epochs) resulted in an AUC of 0.747, sensitivity of 0.753, precision of 0.548, and F1-score of 0.617. Although the adaptive-epoch approach demonstrated superior performance at a 90% masking ratio, the overall ensemble results favored the fixed-epoch approach, suggesting that a balanced distribution of training epochs across all stages contributes to more stable and effective learning.

In addition to the overall results, we present per-class classification performance of CurriMAE in Table 6 to better understand its behavior across individual disease categories. Among the six classes, pneumonia achieved the highest performance, with an AUC of 0.834, sensitivity of 0.879, precision of 0.924, and F1-score of 0.898. This was followed closely by broncho-pneumonia and bronchiolitis, both of which showed strong classification performance across all metrics. The no finding class exhibited moderate

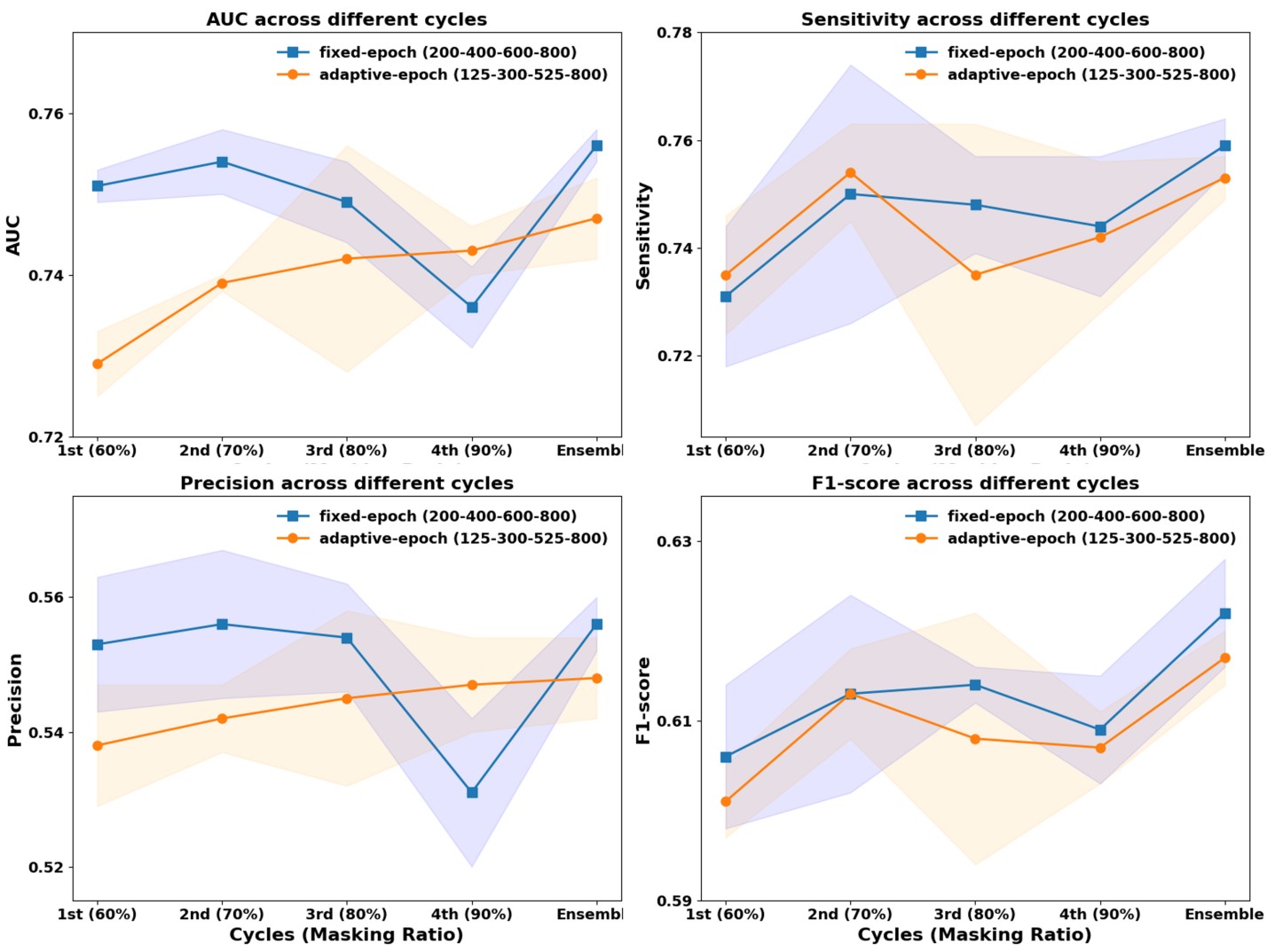

**Figure 3** Performance comparison of fixed-epoch and adaptive-epoch training strategies for multi-labeled pediatric thoracic disease classification.

performance with an AUC of 0.761 and F1-score of 0.724. The other disease class, which includes rare and low-sample conditions, had relatively lower AUC (0.664) and sensitivity (0.681), reflecting the difficulty of learning from infrequent or heterogeneous labels. These results confirm that CurriMAE is capable of handling disease-specific variation in performance while maintaining overall robustness in a multi-label setting.

To enhance the interpretability of CurriMAE's predictions and provide qualitative insight into its decision-making process, we visualized both correctly classified and misclassified examples from the PediCXR test set using attention maps, as shown in Fig. 4. The first row of Fig. 4 displays ground truth images for representative cases, including (A) pneumonia, (B) bronchitis & other diseases, and (C, D) bronchitis & broncho-pneumonia. The second row presents the corresponding CurriMAE attention maps and predicted labels for each case: (E) pneumonia and (F) bronchitis & other diseases, which

**Table 6 Per-class classification performance for CurriMAE on the multi-labeled pediatric thoracic disease task.**

|  | AUC | Sensitivity | Precision | F1-score |
|---|---|---|---|---|
| No finding | 0.761 (0.002) | 0.739 (0.002) | 0.732 (0.002) | 0.724 (0.008) |
| Bronchitis | 0.720 (0.005) | 0.701 (0.012) | 0.841 (0.003) | 0.748 (0.009) |
| Broncho-pneumonia | 0.819 (0.005) | 0.838 (0.012) | 0.921 (0.002) | 0.872 (0.007) |
| Bronchiolitis | 0.712 (0.005) | 0.804 (0.021) | 0.900 (0.003) | 0.844 (0.013) |
| Pneumonia | 0.834 (0.012) | 0.879 (0.011) | 0.924 (0.003) | 0.898 (0.008) |
| Other disease | 0.664 (0.008) | 0.681 (0.046) | 0.908 (0.005) | 0.765 (0.032) |

were correctly classified, and (G) bronchitis and (H) bronchitis, broncho-pneumonia, bronchiolitis, other diseases, which were misclassified. These visualizations demonstrate that CurriMAE can highlight clinically relevant regions associated with its predictions, even in challenging multi-label scenarios.

## DISCUSSION

This study presents CurriMAE, a curriculum-based MAE framework designed to progressively increase masking ratios during pretraining to enhance representation learning while reducing computational cost. The experimental results demonstrate that CurriMAE outperforms standard MAE, ViT-S, and ResNet-34 models in the multi-labeled pediatric thoracic disease classification task, achieving the highest AUC, sensitivity, precision, and F1-score. These findings validate the effectiveness of the progressive masking strategy combined with a cyclic cosine learning rate scheduler, enabling efficient self-supervised learning in medical imaging.

Several previous studies have applied curriculum learning to MIM, similar to CurriMAE. One such approach is CL-MAE (*Madan et al., 2024*), which, like CurriMAE, incorporates curriculum learning into MAE to enhance self-supervised representation learning by progressively increasing task difficulty. Both methods share the fundamental principle that gradually exposing the model to increasingly challenging tasks improves learning efficiency and generalization. They address the critical challenge of selecting an optimal masking ratio, a key factor in MAE training, and both demonstrate improved performance over standard MAE across various tasks. However, their implementations differ significantly. CurriMAE follows a predefined progressive masking schedule, systematically increasing the masking ratio at fixed intervals (60%, 70%, 80%, and 90%) while utilizing a cyclic cosine learning rate scheduler to ensure stable training at each stage. This structured strategy simplifies the training process and enhances learning stability. In contrast, CL-MAE employs a learnable masking module that dynamically adjusts masking complexity based on a self-paced learning strategy. Initially, the module generates easier tasks to facilitate early-stage learning, but as training progresses, it introduces increasingly difficult tasks by optimizing a curriculum loss function that shifts from cooperative to adversarial learning. While this adaptive mechanism enables CL-MAE to generalize effectively across diverse datasets, it also introduces additional computational overhead due to the need for jointly training the masking module. The primary distinction between

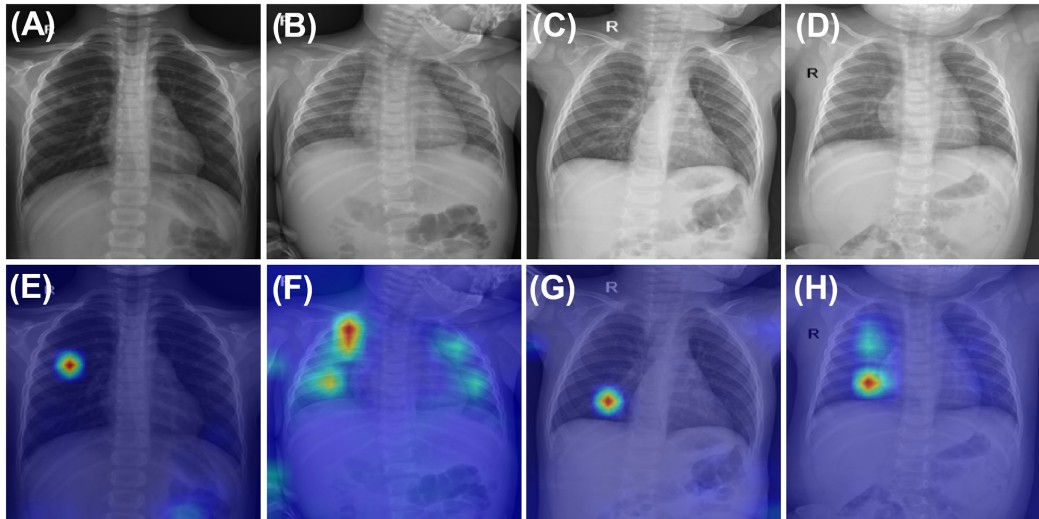

**Figure 4 Attention maps and sample images illustrating correct and incorrect classifications by CurriMAE on the PediCXR dataset.** The first row shows ground truth images: (A) Pneumonia. (B) Bronchitis & Other diseases. (C) Bronchitis & Broncho-pneumonia, and (D) Bronchitis & Broncho-pneumonia. The second row displays the corresponding CurriMAE attention maps and predicted labels: (E) Pneumonia (correctly classified). (F) Bronchitis & Other diseases (correctly classified). (G) Bronchitis (misclassified), and (H) Bronchitis, Broncho-pneumonia, Bronchiolitis, Other diseases (misclassified).

these approaches lies in the structured *vs*. dynamic nature of their curriculum learning strategies.

Another related approach is the prototype-driven curriculum learning framework for MIM proposed by *Lin et al. (2024)*. While CurriMAE progressively increases the masking ratio in stages from 60% to 90% over the course of training, allowing the model to gradually adapt to more difficult reconstruction tasks, the prototype-driven curriculum learning framework begins with simpler, prototypical examples and progressively expands to more complex and diverse instances within the dataset using a temperature-annealed sampling strategy. The key difference lies in the focus of curriculum learning. CurriMAE applies curriculum learning at the masking level, progressively challenging the model by increasing the proportion of masked patches while maintaining the dataset unchanged. In contrast, the prototype-driven curriculum learning framework structures learning at the data level, first training the model on visually representative prototypes and then gradually introducing greater diversity in the training samples. This approach is designed to mitigate early-stage optimization difficulties by preventing the model from being overwhelmed by highly complex samples at the beginning of training. A notable difference also exists in training complexity. CurriMAE generates multiple snapshot models at different masking ratios, requiring separate fine-tuning for each, which increases computational costs during fine-tuning. In contrast, the prototype-driven approach relies on clustering and temperature-based sampling, introducing additional preprocessing overhead but eliminating the need for multiple pretraining snapshots.

Although CurriMAE enhances learning efficiency through progressive masking and reduces the need for exhaustive masking ratio tuning, this study has certain limitations. First, the predefined masking schedule may not be optimal for all tasks, and a dynamic masking strategy that adjusts based on loss convergence could further improve performance. Additionally, this study focused on pediatric thoracic disease classification, and further validation across other medical imaging domains (*e.g.*, MRI, CT) and general computer vision tasks is necessary. Another limitation is that fine-tuning is required for each snapshot model generated during pretraining. In this study, four snapshots were produced, requiring four separate fine-tuning processes, which increases the computational burden in the fine-tuning stage. However, compared to pretraining multiple models with different masking ratios, fine-tuning multiple snapshots is still computationally more feasible. For example, in this study, pretraining required training 303,349 images over 800 epochs, which is significantly more resource-intensive than fine-tuning 7,728 images for 75 epochs. To mitigate the fine-tuning burden in future work, we aim to explore alternative methods for aggregating snapshot models without individually fine-tuning each one, such as employing techniques like model soups (*Wortsman et al., 2022*), which combines multiple models without additional fine-tuning. Last limitation of this study is the lack of access to patient-level metadata such as ICU status, or disease severity in the PediCXR dataset. These factors may affect anatomical presentation and, consequently, model performance. Future work should examine stratified performance when such data become available.

## CONCLUSIONS

This study introduces CurriMAE, a novel curriculum-based pretraining framework that progressively adjusts masking ratios to improve learning efficiency in MAEs. By combining progressive masking with cyclic cosine learning rate scheduling, CurriMAE achieves higher classification performance while mitigating the computational burden of manually tuning masking ratios.

Our experimental results demonstrate that CurriMAE yielded the highest performance among all evaluated models, including MAE, ViT-S, and ResNet-34, for the multi-labeled pediatric thoracic disease classification task. Specifically, CurriMAE attained an AUC of 0.756, sensitivity of 0.759, precision of 0.556, and an F1-score of 0.622, demonstrating its effectiveness, particularly with imbalanced data. Furthermore, we investigated the impact of different epoch allocation strategies during CurriMAE's pretraining. The adaptive-epoch strategy demonstrated significant advantages at higher masking ratios, while the fixed-epoch approach showed more stable results. Ensemble models further enhanced classification performance, validating the benefits of multi-stage training.

Although CurriMAE effectively improves SSL, future research will focus on adaptive masking policies and broader validation in medical imaging and other high-dimensional datasets. These findings contribute to the advancement of curriculum learning in SSL in vision tasks, paving the way for more efficient and scalable deep learning applications. By bridging the gap between computational efficiency and performance, CurriMAE not only

enhances the practicality of MAE but also offers insights into scalable and effective SSL techniques for a wide range of applications.

### Funding

This research was supported by the National Research Foundation of Korea (NRF) through a grant funded by the Korean government (MSIT) under Grant No. RS-2023-00249104. The funders had no role in study design, data collection and analysis, decision to publish, or preparation of the manuscript.

### Grant Disclosures

The following grant information was disclosed by the authors:
National Research Foundation of Korea (NRF).
Korean government (MSIT): RS-2023-00249104.

### Competing Interests

The authors declare that they have no competing interests.

### Author Contributions

- Taeyoung Yoon performed the experiments, analyzed the data, performed the computation work, prepared figures and/or tables, and approved the final draft.
- Daesung Kang conceived and designed the experiments, analyzed the data, prepared figures and/or tables, authored or reviewed drafts of the article, and approved the final draft.

### Data Availability

The CheXpert dataset is available at https://stanfordmlgroup.github.io/competitions/chexpert.

The ChestX-ray14 dataset is available at https://nihcc.app.box.com/v/ChestXray-NIHCC.

The PediCXR dataset is available at https://doi.org/10.13026/xkdz-2d35.

The code for CurriMAE is available at: https://github.com/xodud5654/CurriMAE.

### Supplemental Information

Supplemental information for this article can be found online at http://dx.doi.org/10.7717/peerj-cs.3019#supplemental-information.

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
