# Peer review of "CurriMAE: curriculum learning based masked autoencoders for multi-labeled pediatric thoracic disease classification"

_PeerJ Computer Science, doi:10.7717/peerj-cs.3019_

## Round 0.1 · original submission · Major Revisions

Dear Authors,

Your paper has been revised. It needs major revisions before being accepted for publication in PEERJ Computational Science. More precisely

1) You must explain how you select the beta weights.

2) You must include the computational time and system/hardware specifications used to demonstrate the claim that CurriMAE significantly reduces the CPU time. Detail the computation overhead results more explicitly. Comparative metrics (e.g., number of parameters, CPU time, memory usage) between curriMAE, ResNet, ViT-S, and standard MAE would validate the claim that curriMAE reduces computational cost. Furthermore, you must provide a table summarizing the environmental setup to enhance reproducibility, including hardware specifications and versions of key software libraries or frameworks.

3) You must provide basic demographic info about the dataset, especially age distribution and disease severity. It would also be helpful to know if some X-rays were obtained from ICU patients. The anatomic structure of a 3-year-old and a 17-year-old can be different and may impact the model's performance. Furthermore, given that the study focuses on multilabel classification, it would be helpful to provide the classification performance for each disease separately.

Reviewer 1 ·

Basic reporting

The manuscript is well-organized and clearly presented, with the proposed method—curriMAE—clearly described. The paper is easy to follow and is written in professional English. To further improve basic reporting quality, the following suggestions are offered:

- In the Introduction, please include a brief overview of prior studies on pediatric thoracic disease classification. This should address what the task entails, why it is important, and what work has previously been done.
- Citation for the cyclic cosine learning rate scheduler should be provided (e.g., Line 98).
- In the Related Works section, a dedicated paragraph discussing pediatric thoracic disease classification, including references to prior work in this area, would provide valuable context.
- A summary table at the end of the Related Works section contrasting your method with prior approaches would help clarify your study’s contributions and distinctions.
- Optimal parameter settings for curriMAE should be presented in a concise table for transparency.

Experimental design

The research question is clearly defined, and the study lies within the scope of the journal. The proposed curriculum-style masking strategy appears to be a meaningful innovation. To further strengthen the experimental design:

- Provide a table summarizing the environmental setup, including hardware specifications and versions of key software libraries or frameworks used, to enhance reproducibility.
- Include a README-style instruction outlining the step-by-step execution of the code. This would greatly support efforts toward reproducibility by other researchers.
- Detail the computation overhead results more explicitly. Comparative metrics (e.g., number of parameters, CPU time, memory usage) between curriMAE, ResNet, ViT-S, and standard MAE would validate the claim that curriMAE reduces computational cost.

Validity of the findings

The results support the stated objectives, but additional comparisons and analyses would help confirm the robustness and novelty of the findings:

- As the datasets (e.g., PediCXR) are publicly available, an ablation study comparing curriMAE against prior works using the same dataset would substantiate performance claims.
- Ensure the conclusions drawn are directly linked to the experimental results and avoid generalizations beyond what the data supports.

Reviewer 2 ·

Basic reporting

-

Experimental design

-

Validity of the findings

-

Additional comments

1. How did the authors select those beta weights? E.g., line 269, 311

2. Did they compare the model performance using different parameter settings?

3. Given the author's claim that CurriMAE significantly reduces computation cost, it would be beneficial to include the computational time and system/hardware specifications used.

4. Please provide basic demographic info of the dataset, especially age distribution and disease severity. It would also be helpful to know if some X-rays were obtained from ICU patients. The anatomic structure of a 3-year-old and a 17-year-old can be different and may impact the model's performance.

5. The authors used different datasets for training/testing. It would be helpful to provide a basic comparison to see if the patient characteristics were consistent or varied significantly.

6. Were any medical devices, e.g., tubes, shown on the X-rays? If so, how did the authors handle them in the analysis?

7. Given that the study focuses on multilabel classification, it would be helpful to provide the classification performance for each disease separately as well.

8. In each patch/image, when there are different predictions, please provide more details on how the model decides and makes the final decision.

9. Can CurriMAE locate the disease areas? Or will it just give an overall answer with or without a disease?

10. It would be helpful to visualize some predicted images, from both misclassified cases and correctly classified cases.

---

## Round 0.2 · accepted · Accept

Dear Author,

Your paper has been accepted for publication in PEERJ Computer Science. Thank you for your fine contribution.

Reviewer 1 ·

Basic reporting

No more comments

Experimental design

No more comments

Validity of the findings

No more comments

Additional comments

No more comments